# Ignorance is Bliss:
# Robust Control via Information Gating

**Manan Tomar**[*]
University of Alberta

**Riashat Islam**
McGill University

**Matthew E. Taylor**
University of Alberta

**Sergey Levine**
University of California, Berkeley

**Philip Bachman**
Microsoft Research Montreal

## Abstract

Informational parsimony provides a useful inductive bias for learning representations that achieve better generalization by being robust to noise and spurious correlations. We propose *information gating* as a way to learn parsimonious representations that identify the minimal information required for a task. When gating information, we can learn to reveal as little information as possible so that a task remains solvable, or hide as little information as possible so that a task becomes unsolvable. We gate information using a differentiable parameterization of the signal-to-noise ratio, which can be applied to arbitrary values in a network, e.g., erasing pixels at the input layer or activations in some intermediate layer. When gating at the input layer, our models learn which visual cues matter for a given task. When gating intermediate layers, our models learn which activations are needed for subsequent stages of computation. We call our approach *InfoGating*. We apply InfoGating to various objectives such as multi-step forward and inverse dynamics models, Q-learning, and behavior cloning, highlighting how InfoGating can naturally help in discarding information not relevant for control. Results show that learning to identify and use minimal information can improve generalization in downstream tasks. Policies based on InfoGating are considerably more robust to irrelevant visual features, leading to improved pretraining and finetuning of RL models.

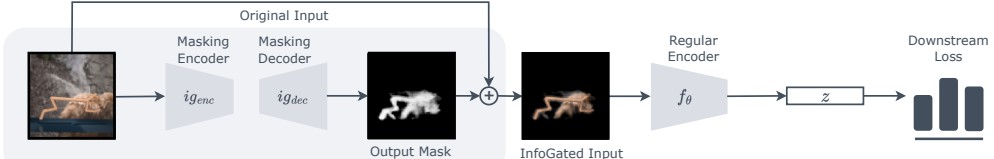

**Figure 1: InfoGating** at the input layer predicts a mask conditioned on the original input and then processes the masked input to compute representations that minimize a downstream loss. The mask is encouraged to remove as much of the original input as possible without hurting performance on the downstream task. The mask is applied by taking a convex combination of the input and Gaussian noise, weighted by the mask.

## 1 Introduction

Pretraining models on large, diverse datasets and transferring their representations to downstream applications is becoming common practice in deep learning. Such representations should capture useful information from available data while ignoring irrelevant features and noise [7, 8]. For instance, an object recognition model may achieve high accuracy on its training set while strongly relying on "spurious" correlations between background features and important objects. When transferred to

---

[*]Correspondence to `manan.tomar@gmail.com`

37th Conference on Neural Information Processing Systems (NeurIPS 2023).

new data with the same objects but a different distribution of backgrounds, performance may suffer. Similarly, a policy for vision-based robot control [23, 10] may perform well in its original training environment, but fail catastrophically when transferred to a new setting [28] where all task-relevant features are the same but minor background features differ from training.

A promising principle for encouraging robustness to out-of-distribution observations is informational parsimony: learning features that contain minimal information [39] about the input while still containing enough information to solve the task(s) of interest. Such regularization is usually based on imposing an *information bottleneck* (IB) [38] on the network that tries to minimize the amount of information flowing from the input, through the bottleneck, to the output prediction. Although IB approaches have been beneficial in some cases, adding an information bottleneck at the penultimate step of computation does little to prevent overfitting in the preceding steps of computation, which typically comprise the overwhelming bulk of the model. To that end, *we consider restricting the information used throughout a computation rather than just restricting the information that comes out of a computation.*

In this paper, we propose learning input-conditioned functions that *gate* the flow of information through a model. For example, we can consider what happens when one inserts an IB near the beginning of a model's computation rather than near the end. In this case, we train the information gating functions to minimize information flow from the input into the rest of the model while still permitting the model to solve the task(s) of interest. We implement this gating using a differentiable parameterization of the signal-to-noise ratio. As the noise level goes up, the signal level goes down, and hence the model displays reduced dependencies on the input. Gating functions can be learnt in conjunction with any downstream loss corresponding to some task of interest. For example, the downstream loss could be a standard contrastive loss for self-supervised learning or an inverse dynamics loss when learning representations for reinforcement learning (RL). In both cases, we can learn gating functions that *reveal minimal information* used to optimize the loss or learn adversarial gating functions that will *remove any information* that can be used to optimize the loss. We primarily focus on InfoGating representations used for learning control policies in RL. A natural focus of the RL paradigm is on learning a mapping from observations to actions, thus discarding most information not useful for control. We bring forth such a notion of only capturing what the agent can affect through InfoGating. Using background distractors and multi-object interactions as a form of noise/irrelevant features, we see that InfoGating is able to remove almost all of the irrelevant information from the pixels, leading to better out-of-distribution generalization in the presence of noise and better in-distribution generalization in the presence of multiple task-irrelevant objects.

Our main contributions are as follows: **1.** A general purpose, practical framework for informational parsimony called InfoGating that can restrict information flow throughout a computation to learn robust representations. **2.** Qualitative analyses on the properties of the gating functions learned with InfoGating that show they are semantically meaningful and enhance intrepretability. **3.** Quantitative analyses of applying InfoGating in the context of various downstream objectives which show clear benefits in terms of improved generalization performance.

## 2   Related Work

**Information Bottleneck**. Much prior work in learning robust representations has stemmed from the idea of imposing an information bottleneck (IB) [38, 39]. Imposing an IB involves maximizing performance on the downstream task, while removing as much information about the input as possible from a network's internal representations. Typical approaches based on Variational IB [2] achieve this by learning stochastic representations that are constrained to be close to a standard Gaussian prior. Variants of dropout [36, 19] can have a similar effect (i.e., adding noise to representations used in the network) but without explicitly maximizing noise/minimizing information in the representations. While IB is typically applied to later stages of a network, InfoGating can be seen as enforcing an IB at the input itself, which is less explored in the IB literature. Like Variational IB methods, and unlike typical dropout-based methods, InfoGating learns how much noise can be added without hurting performance on the downstream task(s). Information Dropout [1] is perhaps the most similar prior work to InfoGating. Information Dropout imposes IBs at multiple points in a network's computation and the IBs are implemented using techniques based on variational dropout [19]. A key distinction between InfoGating and Information Dropout is that InfoGating works post-hoc. With InfoGating a model can consider a computation it has already performed and predict ways in which it could produce the same result more parsimoniously.

**Masked Image Modelling**. There has been a lot of recent work in self-supervised learning where the input is masked and the model is tasked with reconstructing the missing bits [3, 15, 41]. Most of these works operate with a fixed, random masking scheme, instead of learning it directly through the downstream loss. The amount of information masked is also fixed *a priori* whereas we seek to mask as much of the input as possible, while still retaining enough information to solve the task. Some related ideas include learning adversarial augmentations for producing new views of an input during contrastive learning [37], while also learning masks that essentially segment the input space [35].

**Representation Learning in Reinforcement Learning**. Prior work on learning representations for RL includes temporally contrastive learning [33, 27], one-step inverse models [32], learning through next state reconstruction [14], bisimulation metrics [9], and many more. Some of these objectives have been shown to be useful in learning robust representations [40], and we see our method as complementary to these approaches since InfoGating works with any downstream loss function, without needing knowledge of specific values like reward, next state, etc. For instance, bisimulation and task-informed abstractions [11] learn to compress the observation space using reward information, while recent work on learning Denoised MDPs [42] aims at regularizing next state reconstruction using a variational objective. On the other hand, InfoGating remains agnostic to choices of downstream objectives.

## 3 Background

This section describes the primary downstream losses we use with InfoGating in this paper, namely an InfoNCE based contrastive loss and a multi-step inverse dynamics loss.

**Mutual Information Estimation via InfoNCE**. Given $\mathbf{x}$ and $\mathbf{z}$ as two random variables, their mutual information can be defined as the decrease in uncertainty when observing $\mathbf{x}$ given $\mathbf{z}$, compared to just observing $\mathbf{x}$: $I(\mathbf{x}, \mathbf{z}) = H(\mathbf{x}) - H(\mathbf{x} \mid \mathbf{z})$, where $H$ is the Shannon entropy. InfoNCE [31], based on Noise Contrastive Estimation [13], computes a lower bound on $I(\mathbf{z}^1, \mathbf{z}^2)$, where $\mathbf{z}^1$ and $\mathbf{z}^2$ are two representations of the input $\mathbf{x}$ produced by some encoder $f(\mathbf{x})$. Specifically, the bound is optimized by discriminating "positive" and "negative" pairs:

$$\mathcal{L}_{\text{InfoNCE}} = \mathbb{E}_{Z^-}\left[\log \frac{e^{\psi(\mathbf{z}^1, \mathbf{z}^2)}}{e^{\psi(\mathbf{z}^1, \mathbf{z}^2)} + \sum_{\mathbf{z}^- \in Z^-} e^{\psi(\mathbf{z}^1, \mathbf{z}^-)}}\right], \tag{1}$$

where $\psi$ is a pairwise, scalar-valued function of the representations and $Z^-$ is a batch of "negative samples". Typically, self-supervised contrastive learning uses data augmentation such as random cropping and color jittering to define two augmented views $(\mathbf{z}^1, \mathbf{z}^2)$ of a given input as "positives" while views $\mathbf{z}^-$ of other inputs are treated as "negatives". The InfoNCE objective then encourages the representations of positive views of $\mathbf{x}$ to be close (i.e., $\psi(\mathbf{z}^1, \mathbf{z}^2)$ is high), while pushing apart the negatives (i.e., $\psi(\mathbf{z}^1, \mathbf{z}^-)$ is low) [4, 5].

**Multi-Step Inverse Dynamics Models**. Multi-step inverse dynamics models [8] predict the action(s) that took an agent from some observation $\mathbf{x}_t$ to some future observation $\mathbf{x}_{t+k}$, attempting to learn a useful representation of the observations. This resembles learning goal-conditioned policies through relabelling future observations as achieved goals. Although the trivial case of $k = 1$ (a one-step model) does not capture long term dependencies, recent work has shown that multi-step models are able to capture more information that may be useful for controlling the agent [22]. Multi-step inverse models can be used to learn representations by simply predicting the actions $(\mathbf{a}_t, ..., \mathbf{a}_{t+k-1})$ conditioned on $\mathbf{x}_t$ and $\mathbf{x}_{t+k}$. Actions can be predicted using a standard max likelihood objective or with a contrastive reconstruction objective which maximizes the InfoNCE-based lower bound on $I((\mathbf{x}_t, \mathbf{x}_{t+k}), (\mathbf{a}_t, ..., \mathbf{a}_{t+k-1}))$. The learning objective in this case looks as follows:

$$\mathcal{L} = \text{InfoNCE}\big((\mathbf{z}_t, \mathbf{z}_{t+k}), \mathbf{y}_{t:t+k-1}\big),$$

where $\mathbf{z}_t$ and $\mathbf{z}_{t+k}$ are representations computed from $\mathbf{x}_t$ and $\mathbf{x}_{t+k}$, and $\mathbf{y}_{t:t+k-1}$ is a representation computed from $(\mathbf{a}_t, ..., \mathbf{a}_{t+k-1})$. The negative samples in this case come from randomly sampling action sequences of the appropriate length from the agent's collected experience. In the simplest case, the representation $\mathbf{y}_{t:t+k-1}$ may depend only on $\mathbf{a}_t$.

# 4 Information Gating

We present two approaches to InfoGating that we describe as *cooperative* and *adversarial*. Cooperative InfoGating involves keeping as little information as possible to obtain good performance on the downstream task. Adversarial InfoGating involves removing as little information as possible to preclude good performance on the downstream task. We include both approaches in this paper since identifying *minimal sufficient information* (cooperative) and identifying *any useful information* (adversarial) may have different affects depending on the downstream loss and the application domain.

The InfoGating approach has two major components. The first is an encoder of the input $\mathbf{x}$, $f(\mathbf{x})$. The second is an information gating function, $ig(\mathbf{x})$ (see Figure 1). In principle, we can gate information passing through any layer in the representation network that computes the encoding $\mathbf{z} = f(\mathbf{x})$. The $ig(\mathbf{x})$ function provides continuous-valued masks (values in $[0, 1]$) that describe where to erase information from the computation graph for $f(\mathbf{x})$. The shape/size of the output of $ig(\mathbf{x})$ will depend on where we want to gate information. For instance, if we wish to gate information in the input pixel space, $ig(x)$ masks are the size of the image. In general, the goal of $ig(\mathbf{x})$ is to erase as much information from $f(\mathbf{x})$ as possible without hurting task performance.

## 4.1 Cooperative InfoGating

We primarily focus on cooperative InfoGating in this paper and we consider two cooperative variants: gating in the input space or in the feature space.

**InfoGating in Input Space**. We apply InfoGating to an input $\mathbf{x}$ by taking a simple convex combination of $\mathbf{x}$ and random Gaussian noise $\epsilon$. The combination weights are given by $ig(\mathbf{x})$:

$$\mathbf{x}^{ig} = ig(\mathbf{x}) \odot \mathbf{x} + (1 - ig(\mathbf{x})) \odot \epsilon, \tag{2}$$

where $\mathbf{x}^{ig}$ denotes the info-gated input and $\odot$ denotes element-wise multiplication. An all-zero mask corresponds to complete noise (i.e., erasing all information from the input), while an all-one mask corresponds to keeping the original input. The function $ig(x)$ is learnt using the same downstream objective that is used to learn $f(\mathbf{x})$. We encourage masks to remove information from the input by minimizing their L1 norm. This tends to produce sparse masks due to properties of the L1 norm [30]. The overall objective for learning with InfoGating is

$$\mathcal{L}_{ig,f} = \mathcal{L}_{\text{task}}\big(f(\mathbf{x}^{ig})\big) + \lambda \, ||ig(\mathbf{x})||_1, \tag{3}$$

where $\mathcal{L}_{\text{task}}$ refers to any objective through which a useful representation $\mathbf{z}$ can be learnt. Note that when doing InfoGating, we minimize the downstream loss for the masked input $\mathbf{x}^{ig}$ (first term), instead of the original input $\mathbf{x}$, while masking out as much of the input as possible through the L1 penalty (second term). The $\lambda$ coefficient is a hyperparameter that controls how much of the input is masked. In principle, any loss function can be used in conjunction with InfoGating and we consider multiple downstream objectives: contrastive learning of dynamic, Q-learning, and behavior cloning. We describe the overall procedure in Algorithm 1, which assumes contrastive multi-step inverse dynamics modeling as the downstream objective.

**InfoGating in Feature Space**. As an alternative to directly gating the input pixels, we can also consider a variant where $f(\mathbf{x})$ is masked instead of the input. Consider the following masking:

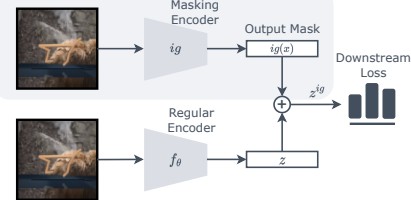

$$\mathbf{z}^{ig} = ig(\mathbf{x}) \odot \mathbf{z} + (1 - ig(\mathbf{x})) \odot \epsilon, \qquad \text{where} \quad \mathbf{z} = f(\mathbf{x}). \tag{4}$$

The training objective remains the same as in the pixel-level case, i.e., minimize the downstream loss $\mathcal{L}_{\text{task}}(\mathbf{z}^{ig})$ for the masked representation $\mathbf{z}^{ig}$, while masking as much of $\mathbf{z}$ as possible (Figure 2). This version is closely related to the deep variational information bottleneck (VIB), which minimizes the KL divergence between the (stochastic) representation $\mathbf{z}$ and a prior distribution, typically chosen to be a standard Gaussian. Roughly, the values in $ig(\mathbf{x})$ can be seen as specifying how many steps of forward diffusion to run in a Gaussian diffusion process initiated at $\mathbf{z}$, where values near zero correspond

**Figure 2: Feature InfoGating**. gates representations produced by the encoder. The gating network and the encoder are jointly trained to minimize the downstream loss, while the gating network is also encouraged to mask as much of $\mathbf{z}$ as possible.

---

**Algorithm 1** InfoGating Pseudocode

---

1: **Input** encoder $f$, masking net $ig$
2: **for** $(\mathbf{x}, \mathbf{x}_{t+k}, \mathbf{a}_t) \in$ loader **do**
3:     $\mathbf{x}_t,\ \mathbf{x}_{t+k} = \text{aug}(\mathbf{x}_t),\ \text{aug}(\mathbf{x}_{t+k})$                                      # random crop augmentation

4:     $\mathbf{x}_t^{ig},\ \mathbf{x}_{t+k}^{ig} = ig(\mathbf{x}_t),\ ig(\mathbf{x}_{t+k})$                                          # get infogates

5:     $\mathbf{x}_t^{ig} = ig(\mathbf{x}_t) \odot \mathbf{x}_t + (1 - ig(\mathbf{x}_t)) \odot \epsilon$                              # infogate current state
6:     $\mathbf{x}_{t+k}^{ig} = ig(\mathbf{x}_{t+k}) \odot \mathbf{x}_{t+k} + (1 - ig(\mathbf{x}_{t+k})) \odot \epsilon$                   # infogate future state

7:     $\mathbf{z}_t,\ \mathbf{z}_{t+k} = f(\mathbf{x}_t^{ig}),\ f(\mathbf{x}_{t+k}^{ig})$                                        # get encodings

8:     **if** Cooperative **then**
9:         Run Adam update on overall loss:
            $\mathcal{L}_{ig,f} = \text{InfoNCE}(\mathbf{z}_t, \mathbf{z}_{t+k}, \mathbf{a}_t) + \lambda\,||ig(\mathbf{x}_t)||_1 + \lambda\,||ig(\mathbf{x}_{t+k})||_1$

10:     **else if** Adversarial **then**
11:         Run Adam update on masking loss:
            $\mathcal{L}_{ig} = -\text{InfoNCE}(\mathbf{z}_t, \mathbf{z}_{t+k}, \mathbf{a}_t) - \lambda\,||ig(\mathbf{x}_t)||_1 - \lambda\,||ig(\mathbf{x}_{t+k})||_1$

12:         Run Adam update on encoder loss:
            $\mathcal{L}_f = \text{InfoNCE}(\mathbf{z}_t, \mathbf{z}_{t+k}, \mathbf{a}_t)$
13:     **end if**
14: **end for**

---

to running more steps of diffusion and thus sampling from a distribution that is closer to a standard Gaussian in terms of KL divergence [18]. Thus, like VIB methods, we aim to minimize $I(\mathbf{x}, \mathbf{z})$, while maximizing the task performance of $\mathbf{z}$. However, feature space InfoGating uses a different optimization objective and parameterization compared to existing VIB approaches. We leave similar variations of InfoGating where arbitrary layers in a computation graph are gated for future work[†].

## 4.2 Adversarial InfoGating

The versions of InfoGating discussed above are cooperative objectives, where the gating network and the encoder work together to lower the overall loss. This leads to finding representations that capture the minimal sufficient information for a given task. We can also turn this formulation on its head, yielding an adversarial objective. In this case, the gating network is tasked with discovering masks that, when used by the encoder, lead to *maximizing* its loss. Instead of encouraging the masks to erase as much of the input as possible, the adversarial objective encourages the masks to remove as little information as possible while minimizing the encoder's performance on the downstream task. We write the adversarial objective for $ig(x)$ and $f(x)$ as:

$$\mathcal{L}_{ig} = -\mathcal{L}_{\text{task}}\big(f(\mathbf{x}^{ig})\big) - \lambda\,||ig(\mathbf{x})||_1, \qquad \mathcal{L}_f = \mathcal{L}_{\text{task}}\big(f(\mathbf{x}^{ig})\big) \tag{5}$$

This gives rise to a min-max objective w.r.t. the encoder and the gating network while cooperative InfoGating corresponds to a min-min objective (see lines 11-12 in Algorithm 1).

## 5 Experiment Setup

To understand whether InfoGating can consistently focus on the minimal possible information required for control, we test generalization performance when using InfoGating with three different downstream control objectives: **1)** contrastive dynamics models (both inverse and forward models), **2)** Q-learning (TD-based critic updates), and **3)** behavior cloning. We choose these three objectives since they remove irrelevant information to varying degrees by default — for example, dynamics models can capture a lot of task-irrelevant information, while behavior cloning models are meant to only contain information that is useful for predicting the optimal policy. Through our experiments, we evaluate whether all three of these objectives can benefit from InfoGating, and to what degree[‡].

---

[†]Constructing differentiable upper bounds on compute cost is one possibility, similar to DARTS [24].
[‡]Since the multi-step inverse model produces the best results, we use the same for testing the feature InfoGating and adversarial InfoGating versions as well.

**Table 1: Multi-step Inverse Dynamics with InfoGating**. Comparing performance in the presence of noisy distractors. We report returns achieved by a policy produced by behavior cloning on top of pretrained representations in *cheetah-run*. Extended results are provided in Appendix C. The "inv" model is our baseline with pretraining via multi-step inverse dynamics. The "w/ Rand" model adds random info gating during pretraining and the "w/ IG" model adds learned info gating during pretraining (this is our method). The easy/medium/hard settings add different levels of distractor noise. Results are for 3 seeds each, with mean and std. dev. reported.

| case | IQL | DRIML | Inv | Inv w/ Dropout | Inv w/ VIB | Inv w/ RCAD | Inv w/Rand | Inv w/ IG |
|---|---|---|---|---|---|---|---|---|
| | | | | | expert | | | |
| easy | $10.7 \pm 8.0$ | $0.9 \pm 0.1$ | $42.6 \pm 36.3$ | $11.79 \pm 3.5$ | $97.1 \pm 17.9$ | $25.0 \pm 3.6$ | $21.0 \pm 15.6$ | $\mathbf{176.2 \pm 9.1}$ |
| medium | $29.2 \pm 28.8$ | $1.0 \pm 0.6$ | $29.5 \pm 31.9$ | $73.3 \pm 12.3$ | $38.9 \pm 16.9$ | $19.9 \pm 24.8$ | $7.2 \pm 6.12$ | $\mathbf{97.0 \pm 5.7}$ |
| hard | $8.8 \pm 1.9$ | $13.8 \pm 7.9$ | $2 \pm 0.6$ | $5.9 \pm 5.7$ | $34.0 \pm 19.4$ | $1.4 \pm 0.4$ | $4.2 \pm 0.7$ | $\mathbf{44.8 \pm 18.4}$ |
| overall | 16.2 | 5.2 | 24.7 | 30.3 | 56.6 | 15.4 | 10.8 | **106.0** |
| | | | | | medium | | | |
| easy | $42.6 \pm 26.4$ | $2.6 \pm 0.2$ | $45.6 \pm 23.2$ | $28.0 \pm 23.6$ | $113.5 \pm 11.8$ | $29.6 \pm 24.5$ | $42.0 \pm 31.8$ | $\mathbf{133.0 \pm 12.8}$ |
| medium | $31.4 \pm 24.7$ | $86.6 \pm 55.0$ | $2 \pm 0.8$ | $39.9 \pm 39.4$ | $44.0 \pm 22.8$ | $4.0 \pm 2.2$ | $85.4 \pm 13.2$ | $92.0 \pm 35.2$ |
| hard | $15.2 \pm 7.7$ | $3.1 \pm 1.5$ | $10.7 \pm 6.0$ | $18.8 \pm 15.2$ | $6.1 \pm 1.8$ | $4.2 \pm 1.6$ | $5.3 \pm 1.9$ | $3.0 \pm 1.43$ |
| overall | 29.7 | 30.7 | 19.4 | 28.9 | 54.5 | 12.6 | 44.2 | **76.0** |
| | | | | | medium-expert | | | |
| easy | $42.4 \pm 31.4$ | $12.9 \pm 12.7$ | $25.5 \pm$ | $45.7 \pm 7.4$ | $130.4 \pm 32.0$ | $51.8 \pm 38.1$ | $10.4 \pm 9.9$ | $158.0 \pm 21.8$ |
| medium | $39.6 \pm 24.7$ | $22.3 \pm 13.6$ | $14.2 \pm 11.2$ | $15.6 \pm 8.1$ | $52.2 \pm 26.0$ | $5.4 \pm 3.9$ | $24.8 \pm 34.4$ | $\mathbf{89.2 \pm 7.0}$ |
| hard | $10.2 \pm 4.0$ | $5.2 \pm 2.6$ | $3.4 \pm 4.8$ | $16.2 \pm 14.4$ | $31.2 \pm 38.7$ | $9.1 \pm 3.4$ | $2.9 \pm 1.2$ | $\mathbf{38.8 \pm 37.5}$ |
| overall | 30.7 | 13.4 | 14.3 | 25.8 | 71.2 | 22.1 | 12.7 | **95.3** |

We test **1)** and **2)** on the offline visual D4RL domain [25] and **3)** on the Kitchen [12] manipulation domain. InfoGating can also be used in non-RL settings like self-supervised learning, as discussed in Appendix A. We now describe the details of each use of InfoGating and the corresponding experimental results. Hyper-parameters are listed in Appendix E.

## 5.1 Inverse Dynamics Models

We use multi-step inverse models as our primary downstream loss due to its ability to recover the latent state effectively [17, 22]. Consider the contrastive loss based on InfoNCE as described in Eq. 1. To apply InfoGating, we mask both the current observation $\mathbf{x}_t$ and the goal observation $\mathbf{x}_{t+k}$ using the masks from $ig(\mathbf{x}_t)$ and $ig(\mathbf{x}_{t+k})$ respectively. We then process both masked observations through the encoder to compute the corresponding representations $\mathbf{z}_t^{ig} = f(\mathbf{x}_t^{ig})$ and $\mathbf{z}_{t+k}^{ig} = f(\mathbf{x}_{t+k}^{ig})$. These are then optimized through a loss similar to Eq. 3:

$$\mathcal{L} = \text{InfoNCE}((\mathbf{z}_t^{ig}, \mathbf{z}_{t+k}^{ig}), \mathbf{a}_t) + \lambda \left( ||ig(\mathbf{x}_t)||_1 + ||ig(\mathbf{x}_{t+k})||_1 \right) \tag{6}$$

Note that both current and future observation masks are penalized through the L1 term. We test the inverse model with InfoGating on datasets consisting of offline observation-action pairings. The datasets contain pixel-based observations with video distractors playing in the background [25]. We first pretrain the representations with the InfoGating objective in Eq. 6 and then perform behavior cloning (BC) using a 2-layer MLP over the frozen representations. We have tried replacing BC-based evaluation with, e.g., Q-Learning-based evaluation, which leads to similar relative scores. We report scores for BC-based evaluation for easier reproducibility and reduced dependence on hyperparameters.

We work with three levels of distractor difficulty: *easy*, *medium*, and *hard*. These levels correspond to the amount of noise added to the observations. At evaluation time, a noise-free observation space is used, thus creating an out-of-distribution shift in the distractor. We use random cropping as a pre-processing step for the observations in all our experiments. We compare inverse dynamics with InfoGating to six baselines: control specific methods, i.e. 1) IQL [21], 2) DRIML [27], and 3) inverse dynamics without InfoGating (the standard formulation from Eq. 1), and regularization specific methods that work on top of inverse dynamics by adding 4) Dropout, 5) VIB bottleneck, 6) adversarial learning (RCAD) [34] and 6) random masks rather than learnable masks as in InfoGating. See Table 16 for results.

We observe that InfoGating leads to considerable performance gains over all baselines. Although adding random masking is better than no masking, learnable masks lead to much better performance.

Since random masking provides similar benefits to data augmentation, some performance improvement over the standard inverse model is expected. However, learning masks clearly does more than just data augmentation. Moreover, InfoGating in the input space also performs more robustly than methods that add regularization in the model weights (Dropout), in an intermediate layer (VIB) or at the output (adversarial samples). This result highlights how valuable removing information in the input space can be for certain applications (as can be visualized in Figure 1). Finally, these results also indicate that minimizing the downstream objective (in this case the inverse model loss is not always sufficient for learning robust representations. By incorporating the InfoGating objective, we add an inductive bias that encourages the model to focus on the most minimal, useful information without additional supervision. This inductive bias towards informational parsimony empirically produces more powerful representations.

## 5.2 InfoGating for Stabilizing Q-Learning

Our prior experiments focused on pretraining representations and then learning regression-based policies, while keeping the representation fixed. In this section, we ask if the same conclusions hold if we train the representation end-to-end with a Q-Learning loss instead. We use the DrQ [20] loss as the downstream objective for this variant of InfoGating. Our InfoGating formulation for Q-Learning essentially amounts to doing standard DrQ training with observations that are masked by the gating network. The gating network learning is driven by the Q-Learning loss directly as follows:

$$\mathcal{L}_{ig,\theta} = \Big( Q_\theta(\mathbf{x}_t^{ig}, \mathbf{a}_t) - \mathbf{r}(\mathbf{x}_t, \mathbf{a}_t) - \gamma Q_\theta'(\mathbf{x}_{t+1}, \mathbf{a}_{t+1}) \Big)^2 + \lambda \, ||ig(\mathbf{x}_t)||_1, \tag{7}$$

where $Q_\theta$ and $Q_\theta'$ denote the current and target Q networks respectively, while $\mathbf{r}(\mathbf{x}_t, \mathbf{a}_t)$ is the obtained reward. In Table 2, we observe that the base DrQ algorithm is quite prone to failure for all three distractor settings, while with InfoGating we see significant improvements in performance. This result shows how using minimal information in the input space can be beneficial in stabilizing TD-based critic training.

**Table 2: InfoGating for Q-Learning**. We use DrQ as the base Q-Learning algorithm and add InfoGating to it. The experimental setup is the same as in Table 16. Results are for an expert policy level.

| case | easy | medium | hard | overall |
|------|------|--------|------|---------|
| DrQ | $5.7 \pm 2.3$ | $29.8 \pm 20.5$ | $3.4 \pm 0.8$ | 12.9 |
| DrQ + IG | $\mathbf{63.4 \pm 22.6}$ | $\mathbf{52.0 \pm 6.2}$ | $5.3 \pm 1.9$ | **40.2** |

## 5.3 Finetuning General Representations with Behavior Cloning

Recent work [29, 26] has investigated learning general representations from large datasets involving diverse object interactions and different varieties of tasks. A natural question arises when using such pretrained representations for downstream tasks — how should the representation be fine-tuned so that only the relevant object and task features for the given task are used for learning the task's policy? InfoGating can be seen as a natural way to extract only information pertaining to the downstream task. Having tested InfoGating extensively on noisy environments, we now study whether there are similar benefits when there is no explicit noise present in the environments, but there are multiple objects/pixel components which could act as potential distractors. We choose five different tasks from the Kitchen [12] environment to test this hypothesis.

Specifically, given pretrained features, we test whether fine-tuning them using InfoGating is more powerful than fine-tuning with only a behavior cloning (BC) loss. We test two pretraining variations here: one corresponding to ImageNet features and the other corresponding to CLIP features. Both are 1) trained on ImageNet data, alongside language clippings for CLIP and 2) fine-tuned using a behavior cloning (BC) policy with a 2-layer MLP attached on top of the pre-trained encoding. Table 3 shows normalized success rates (each out of a maximum score of 100) for both BC and BC with InfoGating. We consistently see that InfoGating is able to mask out most pixel-level information except the robot gripper and the objects being manipulated (see Figure 3 for visualizations of the learnt masks), thus leading to strictly better success rates than the baseline BC policy.

## 6 Ablations

We study various design choices involved in InfoGating and include the most important ablations in this section. Further ablation results are provided in Appendix D.

**Table 3: Finetuning with Behavior Cloning on Kitchen-v3**. We compare fine-tuning performance of BC with and without InfoGating. Representations are pre-trained either on ImageNet labels or through CLIP training. In both cases, InfoGating learns to remove irrelevant objects in the environment surroundings and leads to consistently higher performance.

| env | ImageNet | | CLIP | |
| --- | --- | --- | --- | --- |
| | BC w/ IG | BC | BC w/ IG | BC |
| knob1-on | **17.6 ± 3.1** | 7.3 ± 4.2 | 13.3 ± 1.49 | 11.3 ± 0.94 |
| ldoor-open | 9 ± 5.3 | 5 ± 1.9 | 11.0 ± 4.8 | 4.6 ± 1.49 |
| light-on | 19.6 ± 7.7 | 11 ± 3.6 | 24.0 ± 12.8 | 11.0 ± 3.0 |
| sdoor-open | **61.6 ± 5.0** | 36.6 ± 16.6 | **61.6 ± 8.7** | 42.0 ± 5.9 |
| micro-open | 13.0 ± 7.8 | 4.3 ± 2.9 | 12.0 ± 5.6 | 5.5 ± 2.1 |
| overall | **24.1** | 12.8 | **24.3** | 14.8 |

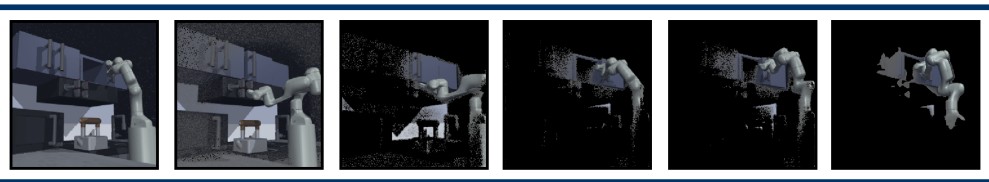

**Figure 3: Visualizing Masks Produced by InfoGating**. Masks improve during training on the Kitchen-sdoor-open-v3 task (left to right corresponds to increasing training steps). InfoGating is able to learn semantically meaningful masks, even beyond the planar control domains such as the cheetah-run task.

**InfoGating in the Feature Space**. We compare the feature InfoGating version with the Variational Information Bottleneck (VIB), while using the same backbone encoders for both methods (see Table 4). Both VIB and feature InfoGating lead to some performance improvement over the base architecture, but do not come close to the score for the pixel InfoGating version. This is potentially due to not removing background distractor information in the first layer itself. Interestingly, feature InfoGating almost matches the pixel InfoGating performance for the hard distractor case. Note that there exist variations such as change in body color, camera zoom, etc in the hard distractor case, that remain even after pixel-level InfoGating. We suspect that feature InfoGating is able to mask out such features but is hurt by the background distractor instead, hence falling to the same performance as the pixel-level variant.

**Table 4: InfoGating in the Feature Space**. We compare the Variational Information Bottleneck with InfoGating in feature space. The experimental setup is the same as in Table 16. Results are for an expert policy level.

| case | easy | medium | hard | overall |
| --- | --- | --- | --- | --- |
| VIB | 97.7 ± 32.2 | 86.2 ± 24.8 | 11.0 ± 5.2 | 64.9 |
| feat. IG | 71.4 ± 44.5 | 76.7 ± 21.3 | **58.9 ± 28.3** | **69.0** |

**Adversarial InfoGating**. We test the adversarial InfoGating algorithm using a multi-step inverse model loss, just as we did for cooperative InfoGating. Figure 4 shows how the adversarial masks tend to hide the agent body. Some additional image content is also erased, since a precise silhouette of the agent's pose would be highly predictive of the agent's pose. To test how useful this kind of masking is, we simultaneously train a separate encoder over the reverse of the adversarial masks. If the adversarial process hides the robot pose successfully, then the reverse mask should contain maximal information about the agent and minimal information about the background. We indeed see that the adversarial InfoGating model performs similarly to the cooperative version, thus verifying that it is an equally viable approach for learning parsimonious representations (see Table 5).

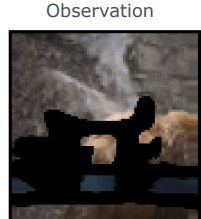

**Figure 4: Left**. Original distractor-based image. **Right**. Learnt adversarial mask over the original image.

**Effect of Mask Sparsity**. We test what range of $\lambda$ values leads to improved performance (Figure 5). As is expected, when $\lambda$ is too high, the entire input is masked and no useful information is captured.

**Table 5: Adversarial InfoGating**. We compare differences in the representations learnt with Cooperative vs Adversarial InfoGating. The experimental setup is the same as in Table 16. Results are for an expert policy level.

| case | easy | medium | hard | overall |
|------|------|--------|------|---------|
| cooperative | $176.2 \pm 9.1$ | $97.0 \pm 5.7$ | $44.8 \pm 18.4$ | **105.9** |
| adversarial | **$202.3 \pm 2.64$** | $84.8 \pm 41.2$ | $4.6 \pm 1.7$ | 96.8 |

Similarly, when $\lambda$ is too low, none of the input is masked and we recover the performance of the base model, one that directly minimizes the downstream loss.

**Mixing Original and Masked Inputs to Encoder**. Since the masking network and the encoder process different kinds of input (unmasked and masked input, respectively), it is worth asking if the encoder can be better regularized by forcing it to be able to process both masked and unmasked images. We can also encourage the encoder to be robust to changes in the input/mask mapping learned by the masking network by occasionally swapping masks between images in a batch while training the encoder. We test both of these separately (Table 6) and observe that training the encoder on masked and unmasked inputs provides strong improvements over the base InfoGating model. This also has the added benefit of not using the masking network at evaluation time, and only using the encoder, which has now been implicitly trained to be invariant to the masked and unmasked inputs. Note that we use this mixed input InfoGating as the default for all our experiments.

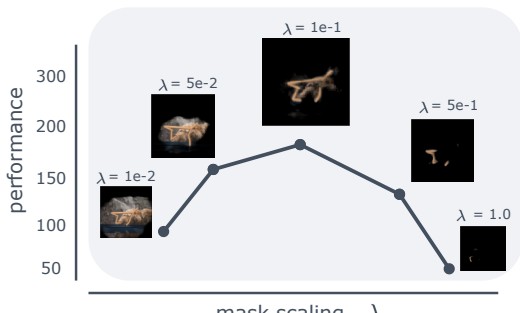

**Figure 5: Scaling** $\lambda$ leads to different masks which reveal different amounts of information. For small $\lambda$, the masked observation still contains distracting/non-salient information, thus hurting performance. Similarly, when masking is too aggressive, too much information is lost and performance goes down.

**Table 6: Mixing Masking and Unmasked Inputs**. Experiment setup follows Table 16. Default InfoGating version is marked in `gray`. Results are for an expert policy level.

| case | easy | medium | hard | overall |
|------|------|--------|------|---------|
| w/ mix input | $176.2 \pm 9.1$ | **$97.0 \pm 5.7$** | **$44.8 \pm 18.4$** | **106** |
| shuffle mask | $170.0 \pm 31.6$ | $54.6 \pm 22.3$ | $9.8 \pm 10.2$ | 78.1 |
| w/o mix input | $119.4 \pm 15.6$ | $18.4 \pm 17.5$ | $10.5 \pm 7.4$ | 49.4 |

## 7 Limitations and Future Work

Although InfoGating helps learn robust representations, it does not recover the same performance for all distractor levels as in the case when no distractors are present. Ideally, we should be able to learn masks that fully remove the background information and thus recover the distractor-free performance. This might be down to two reasons. First, the masks may leave room for noise information to escape around the edges of the object/agent they mask. Second, the encoder that processes the masked images should be robust to slight variations in the masking patterns between training and evaluation samples. In practice, using a UNet for more accurate masks and training the encoder on a mix of masked and unmasked inputs helped remedy these issues quite well, but there still remains room for improvement. This motivates the idea of efficiently InfoGating at multiple layers, without having separate masking networks for each InfoGating layer.

Furthermore, our initial exploration of InfoGating can be viewed as a step towards learning object-centric representations without any explicit notion of objects forced into the model architecture or learning objective. For example, in settings where an agent's actions may affect multiple objects, and we want to predict future observations conditioned on the agent's actions, the masks produced by InfoGating will need to reveal some information about (i.e., "look at") each object. The pursuit of informational parsimony should discourage masks from revealing more of each object than necessary.

## 8 Conclusion

We present InfoGating, a wide-ranging method to learn parsimonious representations that are robust to irrelevant features and noise. We describe two different approaches to InfoGating: cooperative and adversarial, while demonstrating that the gating can be learnt both at the input space or any intermediate feature space. We apply InfoGating to multiple downstream objectives including multi-step inverse dynamics models, Q-Learning, and behavior cloning. InfoGating produces semantically meaningful masks that improve interpretability, and leads to consistently better performing representations, in terms of both out-of-distribution (visual D4RL distractor noise) and in-distribution (irrelevant Kitchen objects) generalization.

## 9 Acknowledgments

The authors would like to thank Charlie Snell, Colin Li, and Dibya Ghosh for providing feedback on an earlier draft, and Katie Kang for suggesting the paper title. Part of this work has taken place in the Intelligent Robot Learning (IRL) Lab at the University of Alberta, which is supported in part by research grants from the Alberta Machine Intelligence Institute (Amii); a Canada CIFAR AI Chair, Amii; Compute Canada; Huawei; Mitacs; and NSERC.

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

# A InfoGating with SimSiam

Random masking has been used as an SSL objective to learn useful features for downstream object detection/classification. Can InfoGating be used as a general strategy to recover similarly useful features? We test this by applying InfoGating with self-supervised representation learning algorithms, in particular SimSiam [6].

The SimSiam [6] objective uses a cosine similarity loss between the representation of one view $\mathbf{z}_1$ and a predicted output of the representation of the second view $\mathbf{p} = p(\mathbf{z}_2)$, where the function $p$ is called the predictor. This is in place of using the InfoNCE loss to define contrastive pairs for $\mathbf{z}_1$ and $\mathbf{z}_2$. Both methods lead to similar performing representations with the main difference being that SimSiam does not require negative examples. The overall SimSiam loss can be written as:

$$\mathcal{L}_{\text{SimSiam}}(\mathbf{z}_1, \mathbf{z}_2) = \mathcal{D}(\mathbf{p}_1, \mathbf{z}_2) \,/\, 2 + \mathcal{D}(\mathbf{p}_2, \mathbf{z}_1) \,/\, 2,$$

where $\mathcal{D}$ denotes the cosine similarity function:

$$\mathcal{D}(\mathbf{p}, \mathbf{z}) = -\frac{\mathbf{p}}{\|\mathbf{p}\|_2} \cdot \frac{\mathbf{z}}{\|\mathbf{z}\|_2},$$

Note that $\mathcal{D}$ is not a symmetric quantity as it uses a stop gradient operation on its second argument $\mathbf{z}$.

Having described how SimSiam works, we can now go into how InfoGating is applied alongside it. Given two augmented views of the input image $\mathbf{x}_1$ and $\mathbf{x}_2$, we mask both views to get $\mathbf{x}_1^{ig}$ and $\mathbf{x}_2^{ig}$, then process them through the encoder to compute the SimSiam loss $\mathcal{L}(\mathbf{z}_1^{ig}, \mathbf{z}_2^{ig})$. Finally, we also add the original SimSiam loss, i.e. one over the unmasked inputs to the overall objective:

$$
\begin{aligned}
\mathcal{L} &= \mathcal{L}_{\text{SimSiam}}(\mathbf{z}_1^{ig}, \mathbf{z}_2^{ig}) + \mathcal{L}_{\text{SimSiam}}(\mathbf{z}_1, \mathbf{z}_2) \\
&+ \lambda \left( ||ig(\mathbf{x}_1)||_1 + ||ig(\mathbf{x}_2)||_1 \right)
\end{aligned}
\tag{8}
$$

We test this version of InfoGating on CIFAR-10, while evaluating performance on Corrupted CIFAR-10 [16]. Figure 6 shows that InfoGating leads to improved evaluation scores in comparison to the base SimSiam method, thus showcasing better robustness to test-time induced corruptions.

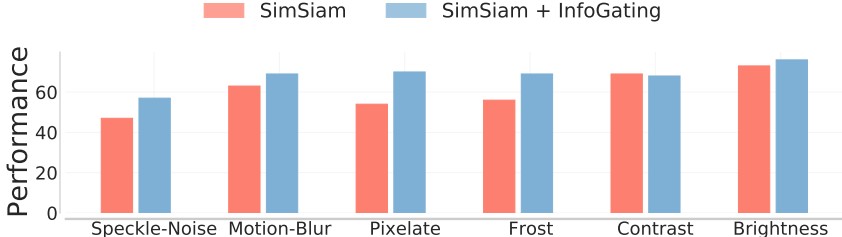

Figure 6: **Generalizing to Corruptions through InfoGating**. We directly compare SimSiam performance w/ and w/o InfoGating on different subsets of the corrupted CIFAR-10 dataset. Adding InfoGating improves robustness to test-time corruptions.

# B Visualizing InfoGating Masks

Learning info-gates directly from the downstream loss allows for adapting the mask based on the task at hand. On the other hand, a fixed random masking scheme offers limited benefits, largely pertaining to enhanced data augmentation. For instance, using an InfoNCE loss derived from augmented views of the input with no distractors leads to learning accurate masks that capture the agent pose almost perfectly (see Figure 7). However, when distractor noise is added, the same InfoNCE loss leads to masks that are spread across the input, failing to remove background noise. Switching the loss to a multi-step inverse dynamics loss leads to masks that focus on the agent pose and successfully remove the background noise. Similarly, using the forward dynamics loss as the downstream objective, masks do not clearly follow the agent pose as in the inverse model case. Similarly, when the

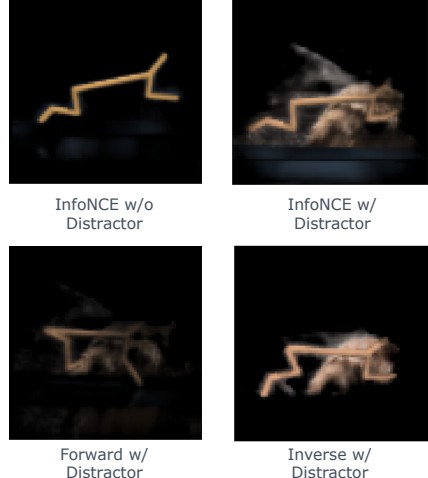

InfoNCE w/o
Distractor

InfoNCE w/
Distractor

Forward w/
Distractor

Inverse w/
Distractor

**Figure 7: Effect of Losses on InfoGating Masks**. InfoNCE w/o Distractor (mean score: **208.0**) learns almost perfect masks, while InfoNCE w/ Distractor (mean score: **10.4**) learns masks that are spread across the input, thus failing to learn a robust enough representation. Forward dynamics w/ Distractor (mean score: **60.4**) and Inverse dynamics w/ Distractor (mean score: **176.2**) lead to much more accurate masks and thus lead to better performing representations as well.

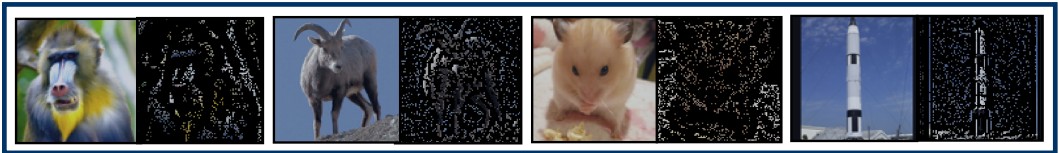

**Figure 8: Visualizing Masks Learnt by InfoGating**. STL-10 images alongside the masked image. The masking network gradually learns to trace the edge boundaries for both foreground and background objects.

downstream objective is a self-supervised loss that focuses on all parts of the input (as opposed to only information useful to control an agent), we see that the learnt masks capture the edge boundaries of both background and foreground objects (see Figure 8). These different variations highlight the importance of choosing the right downstream loss as well as *learning* InfoGating masks in conjunction with the given loss.

## C   Extended Results

Throughout the paper, we use a contrastive variant of the multi-step inverse dynamics loss. This is implemented by first encoding the current and future/goal observations (which are info-gated) $\mathbf{x}_t^{ig}$ and $\mathbf{x}_{t+k}^{ig}$ into corresponding encodings $\mathbf{z}_t^{ig}$ and $\mathbf{z}_{t+k}^{ig}$. We then concatenate the two encodings along with the action to form a single encoding triplet. This is then passed through a 2-layer MLP to output logits for the energy value corresponding to the given $(\mathbf{z}_t^{ig}, \mathbf{z}_{t+k}^{ig}, \mathbf{a}_t)$ triplet. The actions actually taken by the agent make up for a 'positive' triplet and the model is trained to output a low energy value. Similarly, $\mathbf{a}_t$ is replaced by random actions $\bar{\mathbf{a}}_t$ to form a 'negative' triplet, for which the model is trained to output a high energy value.

We can consider a similar version that uses a forward contrastive loss instead. To that end, we simply encode the current action and observation pair $(\mathbf{x}_t^{ig}, \mathbf{a}_t)$ to $\bar{\mathbf{z}}_t^{ig}$) while encoding the future/goal observation $\mathbf{x}_t^{ig}$ to a similar size vector $\mathbf{z}_{t+k}^{ig}$. We then train these by applying standard InfoNCE over the two encodings. The forward model is usually not enough to encode sufficient information for control and therefore lacks the guarantees that a multi-step inverse model has. We test this by info-gating both the current and future observation pair, just as in the inverse model. Next we process both of the masked observations through the encoder, and then concatenate the action $\mathbf{a}_t$ with the $\mathbf{z}_t^{ig}$

representation, before projecting it to the same dimension as the goal representation $\mathbf{z}^{ig}_{t+k}$. Finally, we penalize the info-gates for both current and goal observations as done previously to obtain the following objective:

$$\mathcal{L} = \text{InfoNCE}(\bar{\mathbf{z}}^{ig}_t, \mathbf{z}^{ig}_{t+k}) + \lambda \left( ||ig(\mathbf{x}_t)||_1 + ||ig(\mathbf{x}_{t+k})||_1 \right), \tag{9}$$

where $\bar{\mathbf{z}}^{ig}_t = g(\mathbf{z}^{ig}_t, \mathbf{a}_t)$ is a projection function that maps the state and action embedding to a space of the same size as $\mathbf{z}^{ig}_t$. Note that in the InfoNCE loss, the negatives now come from sampling different future observation representations from the batch.

**Table 7: InfoGating with Forward Dyanmics Models**. Experiment setup follows Table 16. "fwd" is the baseline forward dynamics model and "fwd + IG" adds InfoGating. For reference, training a forward dynamics model without distractors has a return of $173.6 \pm 13.5$.

| case | easy | medium | hard | overall |
|---|---|---|---|---|
| fwd | $6.0 \pm 4.6$ | $7.6 \pm 5.9$ | $4.2 \pm 1.6$ | 5.9 |
| fwd + IG | $\mathbf{60.4 \pm 27.9}$ | $\mathbf{62.1 \pm 29.0}$ | $\mathbf{18.9 \pm 5.2}$ | **47.1** |

# D    Extended Ablations

**Sharing Parameters for Masking Network and Encoder**. Instead of using two different encoders for the masking network and the encoding network, we ask if the same kind of masks can be recovered when sharing parameters between the two encoders (see Table 8). In such a case, the network first outputs the mask by processing the original image through its encoder, then masks the image, and then passes the masked image through its encoder again to generate an embedding vector. Such a shared parameter setup by default ensures that the encoder has a chance to see both masked and unmasked images, which could help avoid overfitting to the distribution of masks output by the mask encoder.

**Table 8: Sharing Masking Network and Encoder Parameters**. Experiment setup follows Table 16. Default InfoGating version is marked in  gray .

| case | easy | medium | hard | overall |
|---|---|---|---|---|
| unshared | $\mathbf{176.2 \pm 9.1}$ | $\mathbf{97.0 \pm 5.7}$ | $44.8 \pm 18.4$ | **106.0** |
| shared | $104.3 \pm 22.6$ | $44.2 \pm 44.0$ | $22.6 \pm 6.3$ | 57.0 |

We generally observe that the performance of InfoGating suffers when sharing parameters. We suspect this result may depend strongly on hyperparameters (e.g. model capacity, noise scale, etc.).

# E    Hyperparameter Details

We use a UNet architecture all throughout this paper for implementing InfoGating on the pixel-level. We use a warm-up consisting of 5k gradient steps where the InfoGating network is not trained, while the downstream encoder is. This is done so that the learnt masks are not affected by initial gradient errors in the downstream loss. Once the warm-up period is over, the loss in Equation 3 is deployed as usual. The details for the masking network (which follows a UNet architecture) and the encoder are described in Table 9 and Table 10 respectively. Note that these network architectures correspond to the visual D4RL locomotion tasks. We add additional layers based on differences in input image sizes (84 x 84) for the Kitchen (256 x 256) and CIFAR/STL-10 experiments (32 x 32 / 96 x 96).

**Table 9: UNet Architecture**.

| Down Sampling (InfoGating Network) |
| --- |
| 3 x 3 conv2d 32, stride 1, pad 1, GroupNorm, ReLU |
| 3 x 3 conv2d 32, stride 1, pad 1, GroupNorm, ReLU |
| 3 x 3 conv2d 64, stride 1, pad 1, GroupNorm, ReLU |
| 3 x 3 conv2d 64, stride 1, pad 1, GroupNorm, ReLU |
| 3 x 3 conv2d 64, stride 1, pad 1, GroupNorm, ReLU |
| MLP |
| FC 128, ReLU |
| FC 128, ReLU |
| FC 1600, ReLU |
| Up Sampling (InfoGating Decoder) |
| 3 x 3 conv2d 64, stride 1, pad 1, GroupNorm, ReLU |
| 3 x 3 conv2d 64, stride 1, pad 1, GroupNorm, ReLU |
| 3 x 3 conv2d 32, stride 1, pad 1, GroupNorm, ReLU |
| 3 x 3 conv2d 32, stride 1, pad 1, GroupNorm, ReLU |
| 3 x 3 conv2d 32, stride 1, pad 1, GroupNorm, ReLU |

**Table 10: Encoder Architecture**.

| |
| --- |
| 6 x 6 conv2d 128, stride 6, pad 0, ReLU |
| 1 x 1 conv2d 128, stride 1, pad 0, ReLU |
| 3 x 3 conv2d 128, stride 1, pad 0, ReLU |
| 4 x 4 conv2d 256, stride 2, pad 0, ReLU |
| FC 256, LayerNorm, ReLU |

# F  Rebuttal

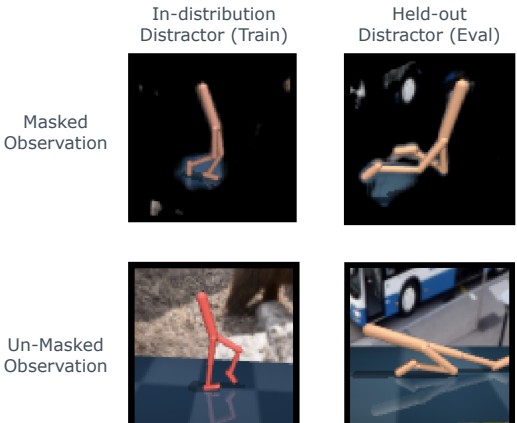

**Figure 9: InfoGating** masks for held-out distractors (during evaluation time), and for in-distribution distractors (during training). The IG masks generalize quite well to unseen distractors.

**Table 11: Visual D4RL Locomotion Training Details**.

| | |
|---|---|
| batch size | 128 |
| $\lambda$ | 0.1 |
| IG warm-up | 5k steps |
| cropping padding | 4 |
| frame_stack | 3 |
| action_repeat | 2 |
| buffer_size | 100000 |
| learning_rate | 1e-4 |
| eval_episodes | 10 |

**Table 12: Kitchen Training Details**.

| | |
|---|---|
| batch size | 32 |
| $\lambda$ | schedule(0.1, 3.0, 2k steps) |
| IG warm-up | 1k steps |
| cropping padding | none |
| frame_stack | 1 |
| num_demos | 5 trajectories |
| learning_rate | 1e-3 |
| IG learning_rate | 1e-4 |
| action_repeat | 1 |
| eval_trajectories | 50 |

**Table 13: CIFAR/STL-10 Training Details**.

| | |
|---|---|
| batch size | 32 |
| momentum | 0.9 |
| weight_decay | 1e-6 |
| $\lambda$ | 0.06 |
| IG warm-up | 100 steps |
| cropping scale | (0.2, 1.0) |
| learning_rate | 0.3 |
| IG learning_rate | 0.3 |
| num_epochs | 60 |

**Table 14: Noisy Evaluation**. Comparing InfoGating performance with multi-step inverse dynamics when training on noisy distractors and evaluating on held-out / out-of-distribution distractors (instead of noise-free evaluation as in Table 1 of the main paper). We also evaluate performance on unmasked observations (right-last column).

| case | IQL | DRIML | Inv w/ Dropout | Inv w/ VIB | Inv w/ RCAD | Inv w/ IG | Inv w/ IG (unmasked eval) |
|---|---|---|---|---|---|---|---|
| | | | | expert | | | |
| easy | $33.8 \pm 6.6$ | $14.0 \pm 5.8$ | $68.4 \pm 12.2$ | $56.9 \pm 1.3$ | $41.9 \pm 11.9$ | $92.3 \pm 19.4$ | $206.4 \pm 5.8$ |
| medium | $12.6 \pm 4.3$ | $12.0 \pm 8.0$ | $28.4 \pm 5.2$ | $29.9 \pm 6.6$ | $13.6 \pm 8.7$ | $48.9 \pm 7.0$ | $60.2 \pm 6.9$ |
| hard | $17.4 \pm 2.9$ | $10.6 \pm 2.1$ | $17.1 \pm 3.9$ | $17.3 \pm 3.9$ | $9.9 \pm 3.8$ | $17.6 \pm 4.5$ | $12.5 \pm 6.4$ |
| | | | | medium | | | |
| easy | $47.5 \pm 12.9$ | $26.1 \pm 0.4$ | $54.4 \pm 4.4$ | $84.2 \pm 10.0$ | $25.1 \pm 12.2$ | $53.1 \pm 7.8$ | $78.0 \pm 27.5$ |
| medium | $13.0 \pm 4.8$ | $16.1 \pm 3.7$ | $22.3 \pm 2.1$ | $18.8 \pm 6.4$ | $5.6 \pm 1.1$ | $48.1 \pm 8.3$ | $130.0 \pm 21.2$ |
| hard | $5.5 \pm 4.8$ | $15.3 \pm 1.6$ | $11.5 \pm 2.8$ | $8.6 \pm 1.7$ | $4.2 \pm 0.5$ | $5.0 \pm 1.0$ | $56.4 \pm 14.7$ |

**Table 15: Walker Results**. Comparing InfoGating performance with multi-step inverse dynamics when training on noisy distractors and evaluating on a held-out distractors (instead of noise-free evaluation as in Table 1).

| case | IQL | DRIML | Inv w/ Dropout | Inv w/ VIB | Inv w/ RCAD | Inv w/ IG |
|---|---|---|---|---|---|---|
| expert | $57.2 \pm 3.9$ | $30.0 \pm 0.7$ | $239.2 \pm 24.7$ | $178.2 \pm 39.3$ | $185.5 \pm 26.9$ | $207.0 \pm 45.0$ |
| medium | $151.0 \pm 24.0$ | $50.0 \pm 6.4$ | $177.3 \pm 28.4$ | $240.0 \pm 30.6$ | $231.4 \pm 12.7$ | $242.0 \pm 20.6$ |
| medium-expert | $135.0 \pm 20.4$ | $57.9 \pm 12.4$ | $210.5 \pm 11.2$ | $187.1 \pm 26.8$ | $207.9 \pm 25.2$ | $215.2 \pm 24.2$ |
| overall | 114.4 | 45.9 | 209.0 | 201.7 | 207.6 | 221.4 |

**Table 16: Reliance on $\lambda$**. Comparing InfoGating performance when using three different $\lambda$ networks *vs* one $\lambda$ network. All experimental settings are the same as in Table 1 of the main paper. For the multiple $\lambda$ version, we do not train on mix of unmasked and masked observations.

| case | Inv w/ IG ($\lambda = 0.1,\ 0.2,\ 0.3$) | Inv w/ IG ($\lambda = 0.1$) |
|---|---|---|
| easy | $165.9 \pm 13.1$ | $176.2 \pm 9.1$ |
| medium | $95.3 \pm 9.3$ | $97.0 \pm 5.7$ |
| hard | $55.6 \pm 5.3$ | $44.8 \pm 18.4$ |

