# OpenReview forum: "Ignorance is Bliss: Robust Control via Information Gating"
_NeurIPS.cc/2023/Conference — NeurIPS 2023 poster_

### Official Review · Reviewer_WnVz · 2023-06-30

**Soundness:** 3 good
**Presentation:** 2 fair
**Contribution:** 3 good
**Rating:** 5
**Confidence:** 4

**Summary:**

This paper empirically investigates a few approaches to modulating the amount of information used by a neural network learner in a variety of control-adjacent learning problems. The proposed approach, InfoGating, learns an input-conditioned continuous-valued mask that is applied to the same input or features thereof. The learner is trained to optimize a task loss and minimize the information allowed through by the mask. Experiments on representation learning via multi-step inverse dynamics modeling demonstrate the relative benefit of InfoGating over other approaches to information parsimony such as the variational information bottleneck. Experiments on Q-learning and fine-tuning pre-trained visual representations for behavior cloning demonstrate the benefit of InfoGating over naive baselines.

**Strengths:**

### Originality
The particular formulation of InfoGating as using input-conditioned masking amortized via a separate neural network is, to my knowledge, original. The authors remark that InfoGating is closely related to previously proposed ideas in the space of information-based regularization of neural networks.

### Quality
The scope of the experiments chosen to validate the proposed idea is perhaps the main strength of this work. Three distinct experimental testbeds (dynamics modelling + probing, Q-learning, fine-tuning pre-trained representations for behavior cloning) are considered, as are a few variants of InfoGating (input gating vs. feature gating, cooperative masking vs. adversarial masking).

### Clarity
The proposed methods are simple and clearly described. However, while some experimental details (architectures and hyperparameters) are presented in the supplementary material, this seems insufficiently detailed to guarantee reproducibility, so I would encourage the authors to release code.

### Significance
The considered problem is important and the proposed methods are simple and intuitively reasonable.

**Weaknesses:**

I list my perceived weaknesses in rough decreasing order of importance.

- (W1) The experimental design leaves a lot to be desired. First and foremost, for the main experiment (multi-step inverse dynamics modeling + behavior cloning probe on half-cheetah), the authors report performance on "a noise-free observation space" (line 221), which I take to mean the original DeepMind Control Suite background or similar. I would instead expect evaluation to use held-out noise (distractions from the Distracting Control Suite) to properly assess the generalization capabilities of the models. As-is, contribution claim 3 (line 62) is unsubstantiated, since, to my knowledge, we typically do not use "generalization" to mean noisy train -> noiseless test (I would instead characterize this as denoising), and neither of the other testbeds include comparisons to other information-based regularization techniques.

- (W2) The design choice of operationalizing information regularization via a learned input-conditioned mask pushes the burden of generalization to the masking network. This seems to be the main distinguishing aspect of InfoGating over other forms of information regularization, yet the impact of this design is neither evaluated in isolation nor in toto.

- (W3) There are several key statements that are unclear or confusing.
    - "Although IB approaches have been beneficial in some cases, adding an information bottleneck at the penultimate step of computation does little to prevent overfitting in the preceding steps of computation, which typically comprise the overwhelming bulk of the model" (line 34). It's not clear to me that IB approaches predominantly bottleneck just before computation output. In particular, methods that use informational/Gaussian/variational dropout seem to contradict this. I would appreciate citations supporting the quoted statements.
    - "Roughly, the values in $ig(x)$ can be seen as specifying how many steps of forward diffusion to run in a Gaussian diffusion process initiated at $z$, where values near zero correspond to running more steps of diffusion and thus sampling from a distribution that is closer to a standard Gaussian in terms of KL divergence" (line 174). This sounds like an interesting connection, but it seems to be only true if the mask specifies a constant value. Otherwise, different pixels are noised to different extents, which doesn't seem to correspond correctly to the standard modelling assumptions in diffusion models.

- (W4) The material in the half-page Background section is completely orthogonal to the proposed idea. I would move a condensed version to the experimental section since the content really is just context for the experiments. Relatedly, Algorithm 1 is InfoGating applied specifically to multi-step inverse dynamics modeling, yet one of the contribution claims is that InfoGating is general purpose.

Addressing W1 and showing good performance on more meaningful out-of-distribution evaluation would cause me to increase my score substantially.


**Questions:**

- (Q1) What do the learned masks look like for i) inputs whose underlying state is trained on, but with held-out distractors; ii) inputs whose underlying state is held-out, but with trained-on distractors; and iii) inputs whose state and distractors are both held-out? This would go a long way in explaining the benefits of (or diagnosing weaknesses with) the proposed approach.

- (Q2) Why does baseline inverse dynamics modeling in Table 1 do so poorly, given the protocol of train-on-noisy, test-on-noise-free? Concretely, why doesn't adding the noise to the training data confer benefits a la data augmentation, since it specifies input transformations that the output (actions) should be invariant to? Is it because the considered noise (frames from continuously playing videos) has temporal consistency?

**Limitations:**

Yes.

---

> ### Author Rebuttal · Authors · 2023-08-10
>
> Thank you for your review and constructive feedback on our paper. We appreciate the time and effort you have put into evaluating our work.
>
> **W1: Evaluations on noiseless environments and W2: Generalizations of masking network**
>
> > Yes, “noise-free observation space” indicates the default DM Control Suite backgrounds. We agree that our main results with background distractors should include results where both training and test episodes include distractors. We performed these tests and present them in Table 1 in the PDF accompanying this rebuttal. We also performed tests in response to your point about offloading generalization duties to the masking network. Specifically, we include results for using the image encoder trained on IG-masked images with unmasked images at test time. Note that this encoder is trained with a mix of masked and unmasked images during training (so unmasked images are not “out of distribution”), and that all other methods use unmasked images (so this comparison is “fair” in this sense).
>
> > From Table 1 in the rebuttal PDF, the image encoder trained via inverse dynamics with InfoGating outperforms the baselines when testing with unmasked images. Depending on how many distractors are seen during training relative to how many are held out for use in testing, we also see some signs of potential overfitting in the mask encoder. I.e., Inverse Dynamics + IG with masked images at test time can suffer when only a small number of distractors are seen during training, even though the same image encoder performs quite well when working with unmasked images during testing.
>
> > We thank the reviewer for pointing out these gaps in our experiments and believe that the expanded results provide stronger support for the claimed contributions in our submission. These results suggest an interpretation of our method as performing task-adaptive data augmentation. I.e. the mask encoder applies cutout to the training images with the goal of cutting out as much as possible without making the task intractable. From this perspective, it is natural to use the IG-trained image encoder with unmasked images at test time, which makes generalization by the mask encoder less critical to our method.
>
> **IB approaches predominantly bottleneck just before**
>
> > We cannot provide a specific citation for our statement that existing IB-like methods predominantly implement bottlenecks at late stages of the overall computation (eg, the last linear layer as in the original work by Alemi et al.), since it is merely an observation on our part. We cite variational information dropout as a counterpoint to our observation, and will edit our claims to be more clear that we’re stating an opinion about how the potential gain from applying some sort of IB early or late in the compute graph is not well reflected in the balance of early/late IB usage in prior publications. We will also clarify that this observation only pertains to “actively info minimizing” forms of IB, and was not intended to include methods based on dropout where the degree of info minimization is set a priori.
>
> **Connections to diffusion models**
>
> > Regarding similarities between InfoGating and diffusion models, we note that the forward diffusion processes in standard DDMs are pixel-wise independent and we posit that one could train a model where the reverse process proceeds at different rates for each pixel. Generating data for training a reverse process with variable per-pixel diffusion rates would be simple due to the pixel-wise independence of the forward process, though one may need some extra bookkeeping to track how far through the forward/reverse process each pixel is. Of course, ease of implementation is not equivalent to ease of getting good results.
>
> **W4: paper structure**
>
> > We can rewrite Algorithm 1 in a more general form in the main paper. We chose to tailor it to inverse dynamics in the submitted draft since we focused a bit more on this setting in the paper.
>
> **Q1: Generalization of IG masks**
>
> > Figure 1 in the rebuttal PDF visualizes the mask for in-distribution and held-out distractors, showing that the the InfoGating masks generalize well to out-of-distribution distractors. Note that since we train on previously collected offline datasets, it is not possible to infer if an underlying state is in-distribution or not.
>
> **Q2: Baseline inverse dynamics performance**
>
> > We include a single, static distractor dataset while training and so your point on "a kind of data augmentation effect coming into play" would be valid if we trained on multiple distractor datasets (creating an implicit invariance to the distractors). The temporal consistency in the background is also a potential reason for the Walker experiments, which uses the single distractor video.
>
> We hope that our rebuttal has addressed your questions and concerns, and we appreciate your consideration in revising the evaluation of our paper.

---

> > ### Comment · Reviewer_WnVz · 2023-08-14
> > **Rating Update**
> >
> > The authors have addressed my main concerns with the experiments. They have added evaluations on held-out distracting backgrounds, showing that their method outperforms prior methods (though not for the "hard" level of noise). They have also shown that a more performant way to use their model at test time is to simply discard the masking network, removing their method's dependence on the masking network's generalization to out-of-distribution inputs. I have increased my rating.

---

### Official Review · Reviewer_TDKb · 2023-07-06

**Soundness:** 3 good
**Presentation:** 3 good
**Contribution:** 2 fair
**Rating:** 6
**Confidence:** 4

**Summary:**

The authors hypothesise that gating information propagation in neural networks will lead to better generalisation. To achieve this they propose a system that performs gates information using a multiplicative differentiable operation, which they call InfoGating. To show the validity of their approach they compare to related models on several tasks such as contrastive dynamics control, Q-Learning and behaviour cloning.

**Strengths:**

1. The method is well motivated.
2. The tasks are relevant to the issue the model is seeking to solve.
3. The comparison models appear to be relevant and well justified.
4. They perform relevant ablations of their design choices.

**Weaknesses:**

While the experimental results show improved generalisation, it is not clear how general this approach will prove to be. So, while I agree that masking irrelevant parts of the input so as to limit distractor information is a good strategy (and one that humans definitely use when interacting with the world), I am less clear that this specific approach fully captures how this process works and will generalise effectively beyond to more complex datasets.

To the author’s credit, they do point out limitations of their approach.

**Questions:**

1. What is the relation between lambda and the overall performance? This is the main hyperparmeter of the system yet there is no graph showing how manipulating it affects performance.
2. If the authors have access to ground truth masks for some or all of the datsets, could they compare the masks produced by their approach with the ones produced by InfoGating?

**Limitations:**

Main technical issue I see is that the reliance on the lambda parameter is very similar to the reliance of a $\beta$ parameter in the $\beta$-VAE literature, which creates a tradeoff that is not always optimal.

It would have been interesting to test the information gating approach with other more principled techniques like object-centric learning which already segment the images into relevant parts (or at least they try to, eg [1]). Then info gating would be only responsible for selecting the appropriate objects for the downstream task.

This approach is also very similar to work in meta-learning which draws on the parallel between gating and neuromodulation in Neuroscience [2]. It would have been interesting to have both a conceptual and experimental comparison between the two.

### References

[1] Locatello, F., Weissenborn, D., Unterthiner, T., Mahendran, A., Heigold, G., Uszkoreit, J., ... & Kipf, T. (2020). Object-centric learning with slot attention. *Advances in Neural Information Processing Systems*, *33*, 11525-11538.

[2] Beaulieu, S., Frati, L., Miconi, T., Lehman, J., Stanley, K. O., Clune, J., & Cheney, N. (2020). Learning to continually learn. *arXiv preprint arXiv:2002.09571*.

---

> ### Author Rebuttal · Authors · 2023-08-10
>
> Thank you for your review and constructive feedback on our paper. We appreciate the time and effort you have put into evaluating our work.
>
> **Question 1: relation between lambda and the overall performance / reliance on the lambda parameter**
>
> > We illustrate the effect of lambda on performance in Figure 5 in the paper which shows how test performance varies with lambda. As lambda increases performance improves until further increasing lambda causes too much information loss. We have added new results in Table 3 of the rebuttal PDF, where we show that InfoGating is considerably robust to $\lambda$ value. Particularly, we trained InfoGating with three masking networks, each with a different lambda value. At each update step, a single masking network is selected at random. We see that the performance is similar to the one obtained when training with a single lambda value, hinting at how InfoGating is not too sensitive to the masking sparsity.
>
> **Question 2: ground truth masks**
>
> > There are no ground truth masks for the settings we consider, and it’s not clear how to produce such masks. However, Figure 7 in the Appendix shows how our masks generally capture the agent pose while removing most other information.
>
> We will note the parallel between gating and neuromodulation in the paper by adding a few lines about the conceptual similarities. Thank you for the pointer.
>
> We hope that our rebuttal has addressed your questions and concerns, and we appreciate your consideration in revising the evaluation of our paper.

---

> > ### Comment · Reviewer_TDKb · 2023-08-14
> >
> > I thank the authors for the reply and additional experiments. My concerns have been addressed and I will update my score accordingly.

---

### Official Review · Reviewer_xdoN · 2023-07-26

**Soundness:** 3 good
**Presentation:** 3 good
**Contribution:** 3 good
**Rating:** 7
**Confidence:** 3

**Summary:**

The authors proposed a mutual information-based encoder that generates masks to gate inputs in order to either pass on minimal information for downstream tasks, or to remove any useful information that could be used to optimize the downstream loss in an adversarial setting. This is a general model that can be applied to a variety of tasks, and the authors focus on the application to reinforcement learning tasks.

**Strengths:**

* Through experiments, the authors show that the idea of directly removing visual information in a learnable way through dynamic masking improves the downstream policy learning in terms of performance and training stability when there are visual distractors.
* InfoGating is very general and could be applied to input or any intermediate representations, with good interpretability


**Weaknesses:**

* training a minimax objective is usually not very stable, and the authors are not very clear in algorithm 1 or the appendix about the details of their minimax training. See questions.
* The idea of masking inputs directly is reasonable, and experiments show that it improves performance and training stability. But it seems not so obviously advantageous on intermediate layers. The authors discussed in Section 6 and 7, but there's no clear statement or experiments towards this line.


**Questions:**

For the minimax objective, are there any nested loops when updating the two sets of parameters?
Is there code available?

**Limitations:**

The authors appropriately addresss ethical or social impacts of their research.

---

> ### Author Rebuttal · Authors · 2023-08-10
>
> Thank you for your review and positive feedback on our paper. We appreciate the time and effort you have put into evaluating our work.
>
> **Minimax objective details**
>
> > We did not use nested loops or different numbers of gradient updates for optimizing the image and mask encoders in the (minimax) adversarial objective. For adversarial losses, we simply switch the sign prior to backprop depending on which parameters we’re updating. We did not encounter stability issues.
>
> **Masking intermediate features**
>
> > We agree that we do not provide strong evidence that masking intermediate features is particularly useful in our RL settings. But, we believe it could be useful in other settings, e.g., for sparsifying attention to reduce compute in LLMs. See, e.g., works like “DARTS: Differentiable Neural Architecture Search” (Liu et al., ICLR 2019).

---

### Official Review · Reviewer_8eSS · 2023-07-27

**Soundness:** 3 good
**Presentation:** 3 good
**Contribution:** 2 fair
**Rating:** 6
**Confidence:** 1

**Summary:**

This paper introduces a novel masking technique called InfoGating for learning masks in contrastive loss settings with InfoNCE.  The proposed approach is simple, well-motivated, and is evaluated in several RL setting including inverse dynamics models, Q-learning, and behavior cloning.  Some ablation studies on regularization of the mask, as well as variants of the objective are also considered.

**Strengths:**

* The proposed approach is a novel way for learning masking and is applied for RL tasks, whereas traditional approaches are typically evaluated for classification, or SSL tasks.  Evaluation of the proposed approach covers multiple RL frameworks including Q learning, behavior cloning, and inverse dynamics.
* The proposed method is stated very clearly including all hyperparameters for reproducibility, clear pseudocode for inverse dynamics, and a clear description of the approach with intuition.

**Weaknesses:**

* My primary concern with this paper is that the proposed approach does not have sufficient evaluation not situate itself well with prior works.  There are two notabl method comparisons, VIB (2017), and RCAD (2022).  Neither of these approaches, however are masking-based approaches which is the main line of this work from my point of view.  These methods are also not included in comparisons on behavioral cloning and Q-learning experiments.  I would like to see additional comparisons on Q-learning and behavioral cloning, and perhaps some experiments comparing other masking techniques which from my understanding have been used for RL such as Q-learning: https://proceedings.neurips.cc/paper_files/paper/2022/hash/a0709efe5139939ab69902884ecad9c1-Abstract-Conference.html, https://proceedings.neurips.cc/paper_files/paper/2022/hash/802a4350ca4fced76b13b8b320af1543-Abstract-Conference.html.  Although these approaches approach masking from a slightly different angle, my understanding is that they are still aimed at better observation learning.  There are also works orthogonal to this that focus on sparse convolutions: https://openaccess.thecvf.com/content_CVPR_2020/html/Verelst_Dynamic_Convolutions_Exploiting_Spatial_Sparsity_for_Faster_Inference_CVPR_2020_paper.html.  I think contrasting with these approaches may also distinguish the paper.
* Additionally, cheetah run is only one of many tasks within the continuous control suite.  It would be good for the authors to clarify why this task in particular, and if IG performs well across multiple additional tasks.  Further, the reported cheetah run numbers are a fair bit lower than those traditionally reported for the return (usually 300+).  Is there commentary on why the numbers are much lower in this setting, and why learning curves are not provided?
* There are some followup questions about whether or not information gating without controlling  for consistency in all layers is good.  In particular, whether this may violate properties of the IB principle which say that the information will only decrease throughout the network.

**Questions:**

* How does infogating satisfy or violate the IB principle in deep learning?  Is it possible for the model to result in higher information in later layersdepending on how much gating happens in features of the network? Does this violate any benefits of the IB principle, for example better generalization?
* How does IG perform compared with other methods for masking/information bottleneck on Q-learning and behavior cloning settings?
* What is the connection between IG and adversarial masking?

**Limitations:**

The proposed approach is not compared with other masking approaches for traditional augmentation and image classification settings.

---

> ### Author Rebuttal · Authors · 2023-08-10
>
> Thank you for your review and constructive feedback on our paper. We appreciate the time and effort you have put into evaluating our work.
>
> **Comparison with Masking-based methods**
> > We ran new tests using mask-based latent reconstruction (MLR). Even after extensive hyper-parameter tuning, we were not able to avoid collapse in representation. While we were able to produce meaningful results after slightly modifying the method, the results in each case were inferior to our method. In our RL setting, our method is best seen as a form of regularization. This is a somewhat different goal from the Dynamic Convolutions work.
>
> **Results beyond Cheetah domain**
> > Our new tests include results in the Walker domain. We primarily focused on Cheetah because the vd4rl domain only included distractor datasets for all three policy levels for Cheetah. For the Walker results, we collected our own distractor datasets. The results are provided in Table 2 in the rebuttal PDF. We largely observe that InfoGating performs better than the other five baselines.
>
> **InfoGating and the IB principle**
> > Our method strictly removes information from the input, and information can’t be added as one progresses through a compute graph (without taking additional inputs) due to the data processing inequality.
>
> **Other IB methods in the Q-learning and Behavior Cloning Setting**
> > Q-learning - Since the Q-learning setup is quite similar to the inverse dynamics, i.e. they both use the same encoder, we expect the dropout results to be similar to that of the inv w/ dropout results in Table 1 of the main paper.
>
> > Behavior Cloning - In our preliminary experiments, we added dropout to the policy network for both behavior cloning and behavior cloning w/ IG versions. We did not see any significant improvement in performance for both versions, while keeping the model architectures fixed. It is quite likely that since the heavy lifting is done by the representation encoder, adding dropout to the policy network does little to improve performance.
>
> > Because of the bottleneck on compute resources, we could not run more experiments to validate these preliminary observations. We are happy to include these in the final version of the paper if the reviewer feels strongly about these results.
>
> **Connection between IG and Adverarial Masking**
>
> > The “cooperative” version of InfoGating is quite different from Adversarial Masking as in ADIOS. In particular, the IG masks in this setting seek to reveal the minimal info required for solving a task. The “adversarial” version of IG is similar to a single-mask version of ADIOS. Extending IG to use multiple masks is straightforward.
>
> We hope that our rebuttal has addressed your questions and concerns, and we appreciate your consideration in revising the evaluation of our paper.

---

> > ### Comment · Reviewer_8eSS · 2023-08-19
> > **Response to Author Rebuttal**
> >
> > Thank you for the response, and addressing many of my concerns with the extra experiments.
> >
> > * Given experiments on other masking approaches are completed, it would be nice to see the final results in the paper (or appendix) as I believe this is an important comparison.
> > * Experiments with the Walker look convincing, and demonstrate the approach generalizes across environments.
> > * My concern with the data processing inequality is with masking. If I denote input at layer $\ell$ as $X_{\ell}$, then data processing inequality says $MI(X_{\ell}; Y) \geq MI(X_{\ell+1}; Y)$.  My question is whether this holds when applying masking i.e. $MI(M(X_{\ell}); Y) \geq MI(M(X_{\ell+1}); Y)$ for the mask function $M$.  A potential violation could happen if there is no constraint on $M$, for example $M$ could mask everything at $\ell$ and nothing at $\ell+1$.
> >
> > Nonetheless, I will update my score to reflect positive results from the updated experiments.

---

### Official Review · Reviewer_L82J · 2023-07-28

**Soundness:** 3 good
**Presentation:** 3 good
**Contribution:** 2 fair
**Rating:** 6
**Confidence:** 3

**Summary:**

In this paper, the authors propose InfoGating as a way to learn parsimonious representation that could achieve better generalization by being robust to noise and spurious correlations. Representations that identify the minimal information required for certain task. Those representations attempt to be robust to out-of-distribution observations.

In contrast with previous works that applied a similar concept, i.e., information bottleneck, in later stage of the computation, InfoGating is applied to the input layer, which allows models to learn which information is key to solve the task. For example, what pixels matter for a given task.

There are two approaches proposed for InfoGating namely cooperative and adversarial. The former aims to identifying minimal sufficient information. The latter to identifying any useful information.


**Strengths:**

S1. The method is simple and well explained, easy to read. It sounds correct.

S2. I found the experimental setup very interesting and well choose. I have seen methods motivated by similar concepts in self-supervised learning and efficient transformer [ATS: Adaptive token sampling for efficient vision transformers. Fayyaz et al. ECCV-22], but not for RL tasks.

S3. Ablation results on feature space vs pixel InfoGating shows the importance of removing distractors early in the pipeline. Supporting the motivation of this work.

S4. Appendix, Visualization and additional results are good and provide good insides to readers.

**Weaknesses:**

W1. Since the InfoGaiting is presented as a general method to obtain more robust representations that can deal with out-of-distribution observations, I was expecting to see experiments in more dedicated settings like NICO++ [NICO++: Towards Better Benchmarking for Domain Generalization]



**Questions:**

Q1. Can you please explain further what is the metric used on the experiments?

Q2. In the experiments with inverse dynamic models, How consistent is the masking between two consecutive frames?



**Limitations:**

I found the limitations are well covered.

---

> ### Author Rebuttal · Authors · 2023-08-10
>
> Thank you for your review and positive feedback on our paper. We appreciate the time and effort you have put into evaluating our work.
>
> **Benchmarking on vision-based datasets**
>
> > Since we focus on RL settings, we did not test on vision-based domain generalization, but this could be interesting in future work. Our new experiments evaluate on multiple domains (multiple background distractors, please see Table 1 in the rebuttal PDF), going beyond the leave-one-out evaluation strategy, as pointed out in the NICO++ paper.
>
> **Q1. Can you please explain further what is the metric used on the experiments?**
>
> > We evaluate using the return obtained after training a linear policy on representations learned with any given method. The representations are trained with distractor-based observations while the policy is evaluated without distractors. Our new experiments show that these conclusions still hold if the policy is evaluated with distractors.
>
> **Q2. In the experiments with inverse dynamic models, How consistent is the masking between two consecutive frames?**
>
> We observe masks to be consistent across consecutive frames. The masking network learns to focus on the robot contours, as shown in Figure 7 in the Appendix. It would be interesting to see what happens when inferring masks simultaneously across multiple frames, since temporal consistency could be leveraged to further reduce the number of visible pixels.
>
> We hope that our rebuttal has addressed your questions and concerns, and we appreciate your consideration in revising the evaluation of our paper.

---

> > ### Comment · Reviewer_L82J · 2023-08-17
> >
> > Thanks to the authors for addressing my questions and concerns. They have added experiments on multiple backgrounds distractors, Table 1, a similar setup of NICO++. I will update my score accordingly.

---

### Author Rebuttal · Authors · 2023-08-10

We thank the reviewers for their feedback. We have individually responded to points made by each reviewer and provided further experiments in support of our response (please refer to the attached rebuttal PDF). Below is a list of additional experiments we have included:

1. **Evaluations of all baselines and InfoGating on multiple distractors (including unmasked evaluations for IG):** We show that even when evaluations are not done on noise-free observations, InfoGating is considerably better performing.

2. **Results on the walker domain, for all baselines and InfoGating.**

3. **Visualizations of InfoGating masks on in-distribution and held-out distractors.**

4. **InfoGating with multiple masking networks**, each with a different $\lambda$ value, showing that IG relies very loosely on the particular  value of $\lambda$.

We hope our response and the additional results can help address the reviewers' questions and concerns. For criticisms not addressed in this rebuttal, we invite the reviewers to point out those which seem most pressing during the interaction period. Thank you!

---

### Decision · Program_Chairs · 2023-09-21

**Decision:**

Accept (poster)

**Comment:**

This paper presents information gating as a mechanism to use minimum task-relevant information to improve robust learning. The method is applied Q-learning. It boosts performance by concentrating on the informative content, enhancing the robustness of the reinforcement learning model.

The reviewers' feedback acknowledges the strong points of the paper, particularly its clear presentation and its application to reinforcement learning. They appreciate the insights from your ablation results and the overall experimental setup. All reviewers recommend unanimously for acceptance.

Whilst, reviewers emphasise the need for more extensive evaluation, suggesting direct comparisons with relevant existing methods and a broader range of tasks to demonstrate generalisability. They've also highlighted the need to discuss the relation of the method with attention.

The AC had a chance to read the paper closely and went through all the reviews. Besides the strong points mentioned above, the paper lacks in that the method is not discussed and compared with attention methods. The proposed InfoGating has an alternating optimisation, but that requires more ablation to be convincing.  It holds true that there may not be a widely recognized practice in the field of RL to incorporate attention for information gating, but the AC strongly encourages the authors to consider these insights for refinement. Additionally, incorporate the experiments from the rebuttal phase in the paper.